# Genomic Signatures of SARS-CoV-2 Associated with Patient Mortality

**DOI:** 10.3390/v13020227

**Published:** 2021-02-02

**Authors:** Eric Dumonteil, Dahlene Fusco, Arnaud Drouin, Claudia Herrera

**Affiliations:** 1Department of Tropical Medicine, Vector-Borne and Infectious Disease Research Center, School of Public Health and Tropical Medicine, Tulane University, New Orleans, LA 70112, USA; dfusco@tulane.edu (D.F.); cherrera@tulane.edu (C.H.); 2Department of Medicine, School of Medicine, Tulane University, New Orleans, LA 70112, USA; adrouin@tulane.edu; 3Department of Pathology, School of Medicine, Tulane University, New Orleans, LA 70112, USA

**Keywords:** COVID-19, coronavirus, pathogenesis, SNP, genome

## Abstract

Infections with SARS-CoV-2 can progress toward multiple clinical outcomes, and the identification of factors associated with disease severity would represent a major advance to guide care and improve prognosis. We tested for associations between SARS-CoV-2 genomic variants from an international cohort of 2508 patients and mortality rates. Findings were validated in a second cohort. Phylogenetic analysis of SARS-CoV-2 genome sequences revealed four well-resolved clades which had significantly different mortality rates, even after adjusting for patient demographic and geographic characteristics. We further identified ten single-nucleotide polymorphisms (SNPs) in the SARS-CoV-2 genome that were associated with patient mortality. Three SNPs remained associated with mortality in a generalized linear model (GLM) that also included patient age, sex, geographic region, and month of sample collection. Multiple SNPs were confirmed in the validation cohort. These SNPs represent targets to assess the mechanisms underlying COVID-19 disease severity and warrant straightforward validation in functional studies.

## 1. Background

In December 2019, an outbreak caused by a novel Coronavirus referred to as SARS-CoV-2 quickly progressed into one of the worst pandemics, causing an unprecedented international health and economic crisis. As of October 2020, there were over 45 million confirmed cases worldwide, with over 1.2 million deaths (corresponding to a global 2.7% mortality rate) according to the Johns Hopkins Coronavirus Resource Center [1]. However, infections with SARS-CoV-2 can progress toward very variable clinical outcomes, ranging from asymptomatic infections to very severe pulmonary disease and death. The identification of host and viral factors associated with disease severity would represent a major advance to guide medical care and improve patient prognosis [2]. It could also allow the identification of precise therapeutic targets such as host pathways or specific viral proteins involved in pathogenesis.

Mortality rates can vary greatly according to geography, population density, demographics, the timing and extent of community mitigation measures, testing availability, healthcare infrastructure, and public health reporting practices, among others [3]. Nonetheless, epidemiologic studies allowed for identifying some host factors associated with a higher mortality, including increased age, male sex, and several chronic comorbidities such as obesity, diabetes, or coronary artery disease [4,5]. An exacerbated immune response in patients together with an impaired type I interferon has also been proposed as a critical contributor to disease severity [6,7].

Viral factors underlying COVID-19 disease severity and mortality are, on the other hand, unclear [6]. Comparison of genome sequences among SARS-CoV-2 and other members of the *Coronaviridae* family infecting humans, including SARS and MERS, indicated that differences in the nucleocapsid (N) and spike (S) viral proteins may be associated with the increased mortality rate caused by these coronaviruses as well as with the host switch from animals to humans [8]. Among SARS-CoV-2, a mutation hot spot was first identified in the virus RNA-dependent RNA polymerase (RdRp), but mutations have also been detected elsewhere throughout the viral genome [9,10], although their relevance for pathogenesis is unknown. The G614D variant in the S glycoprotein has been associated with increased transmissibility, infectivity and viral loads, but not with disease severity [11,12]. Similarly, a G/T variant in the Open reading frame 1ab (Orf1ab) gene has been associated with symptomatic and asymptomatic infections, respectively, in a small cohort of 152 patients [13], and some viral clades from Chicago have been associated with differences in viral loads [14]. Another study found correlations between mortality rates at the country level and the frequencies of G614D and P4715L variants in the S and Orf1ab proteins, respectively [15]. In this study, we aimed to identify genomic signatures of SARS-CoV-2 virus that may be associated with mortality of infected patients. Therefore, we tested for potential associations among viral genomic variants from a large international cohort of 2508 patients and their clinical and demographic characteristics.

## 2. Methods

### 2.1. Viral Sequence Data and Associated Patient Metadata

A dataset of 3205 whole-genome sequences from SARS-CoV-2 virus was selected from the Global Initiative on Sharing All Influenza Data (GISAID) database (https://www.gisaid.org), based on the availability of links with patient metadata, including disease severity and demographics (Appendix A). These sequences corresponded to virus isolates from multiple regions, including Asia, Africa, Europe, Oceania, and America, collected between December 2019 and 26 June 2020, from over 402 laboratories.

The cohort included both inpatients and outpatients, and most sequences were derived from oro- or naso-pharyngeal swabs (1691 sequences), secretions/sputum (105 sequences), broncho-alveolar fluid (42 sequences), and other or unspecified samples. No information on patient ethnicity was available, nor on potential co-morbidities. Because many of the clinical terms used to describe disease status were ambiguous/vague and to avoid any bias in categorizing disease severity, we only focused on mortality versus survival (case fatality rate). We considered patients described as “Deceased” as dead, while those described as “Alive, Asymptomatic, Cured, Discharged, Discharged after recovery, Facility quarantine, Fever, Home, Hospitalized, Intensive Care Unit, Severe, In hospital, Inpatient, Isolation, Live, Mild, Moderate, Outpatient, Not Hospitalized, Quarantine(d), Recovered, Released, Symptomatic” were considered as survivors. After deleting incomplete/low-quality sequences, and samples with inconsistent metadata, a final dataset of 2508 sequences with associated patient mortality, collection date, geographic region, patient age, and sex was used for analysis.

We also obtained a second independent dataset from GISAID for validation of our findings. It consisted of 1488 new viral sequences deposited between June 27 and 23 July 2020 (Appendix A). These included virus sequences from multiple regions, as above, collected between 1 January and 23 July 2020. After deleting incomplete/low-quality sequences, and samples with inconsistent/missing metadata, a validation dataset of 992 sequences with associated patient mortality, collection date, geographic region, patient age, and sex was used for analysis.

### 2.2. Data Analysis

Viral genome sequences from our first dataset were aligned using MAFFT [16] as implemented in Geneious 11, and a phylogenetic tree was built using FastTree, which infers approximately-maximum-likelihood phylogenetic trees [17]. Major clades were compared to GISAID clades for easier reference. We tested for potential association of viral clades with mortality rates using X^2^ tests. Mortality rates were also compared according to demographic, geographic and temporal data to assess potential confounders. We used X^2^ tests to assess differences in mortality rates according to sex, month of year samples were collected and geographic region. We also compared the age of deceased and survivor patients using *t*-test, and compared patient age among viral Clades through Tukey post hoc test. No specific analyses could be performed associating mortality with ethnicity or preexisting comorbidities, as these were not reported in these datasets. Next, we used Generalized Linear Models (GLM) to test for association of mortality rates with viral clades while adjusting for demographic, geographic and temporal parameters, assuming a binomial distribution and a Logit link function with Firth Bias-adjusted estimates. Sequences from Oceania were excluded from this analysis due to a small sample size of sequences from this region (15 sequences), which caused model instability.

To identify genomic variants, Single-Nucleotide Polymorphisms (SNPs) were called from SARS-CoV-2 genome alignment through Geneious 11 SNP/variant tool, and tested individually for association with mortality, using X^2^ tests and odd ratios (OR). Statistical significance was adjusted using Bonferroni correction to account for multiple testing. SNPs positions in the genome were determined based on a sequence from Wuhan, China, from December 30, 2019 (Genbank # MT291827) as reference. We also assessed the phenotypic effect of each SNP in the corresponding viral protein. Finally, we analyzed combinations of SNPs found significantly associated with mortality in the bivariate analysis and further adjusted for demographic and geographic covariates in a multivariate model based on a GLM as described above. Again, sequences from Oceania were excluded from this analysis due to insufficient sample size from this region. We elaborated several models with different combinations of SNPs and covariates and models were compared based on Akaike Information criteria (AICc) to select the best model. All statistical analyses were performed in JMP 9.0.

### 2.3. Analysis of Validation Dataset

SARS-CoV-2 genome sequences were aligned as described above, and nine of the SNPs associated with mortality in the first cohort of patients were identified in this second cohort and their nucleotide variants scored. SNP association with mortality was tested by X^2^ tests, and statistical significance was adjusted using Bonferroni correction to account for multiple testing, as above. Finally, we also analyzed combinations of SNPs found significantly associated with mortality in the bivariate analysis and adjusted for demographic and geographic covariates in a multivariate model based on a GLM.

## 3. Results

Phylogenetic analysis of SARS-CoV-2 genome sequences from our cohort revealed four well-resolved clades (Figure 1). The overall mortality rate in this cohort was 5.74% (144/2508). However, patient mortality varied significantly among the identified clades (X^2^ = 47.93, d.f. = 3, *p* < 0.0001), ranging from 2.06% [95%CI 1.28–3.32] for Clade 1 up to 11.61% [95%CI 8.97–14.91] for Clade 3, with Clades 2 and 4 having intermediate mortality rates (6.03% [95%CI 4.35–8.31] and 5.84% [95%CI 4.34–7.83], respectively). These results suggested an association of viral clades with mortality.

However, deceased patients were also of older age (66.8 ± 1.31 vs. 48.2 ± 0.4 years for survivors, t-test, *p* < 0.0001), and mortality rates varied significantly according to the geographic region (X^2^ = 61.26, d.f.= 4, *p* < 0.0001), time of year (X^2^ = 74.35, d.f.= 6, *p* < 0.0001), and tended to be higher in males (X^2^ = 2.37, d.f. = 1, *p* = 0.12) (Appendix A). Therefore, we adjusted for these variables in a Generalized Linear Model (GLM), which confirmed that patient mortality was significantly associated with SARS-CoV-2 clades (Effect test *p* < 0.0001), together with patient age (*p* < 0.0001), geographic region (*p* < 0.0001), time of year (*p* < 0.0001), and sex (*p* = 0.004) (Appendix A). The sex-ratio of infections was similarly biased toward more males for all clades (X^2^ = 4.02, d.f. = 3, *p* = 0.25, Figure 2A). Patient age distributions were similar among the Clades, although the mean age was significantly lower for Clade 1 (*p* < 0.01, Figure 2B). There was a significant difference in the proportion of the respective clades among geographic regions (X^2^ = 522.65, d.f. = 12, *p* < 0.0001), and Clade 1, associated with a lower mortality rate, was predominant in Asia, and Clade 4 was predominant in North America (Figure 2C). The proportion of each Clade also varied over time (X^2^ = 821.54, d.f. = 18, *p* < 0.0001) (Figure 2D). Initial infections were caused by virus from Clade 1, which started to be replaced by the other Clades in February 2020, and became nearly absent from this cohort by June 2020 (1.45% of all sequences). Clade 3, associated with a high mortality rate, presented an initial increase in proportion in March and April 2020, but decreased since then, and Clade 2 and 4 were the most frequent clades in the cohort as of June 2020 (Figure 2D). However, there were regional variations in the changes in Clade proportions over time, and the only constant observation was the progressive replacement of Clade 1 (Figure 3).

We then focused on identifying specific sequence variants underlying these differences in mortality rates among SARS-CoV-2 clades. Viral genome sequences were analyzed for Single-Nucleotide Polymorphisms (SNPs), and we identified a total of 27 positions with SNPs ranging in frequency from 2.4 to 68.1% (Table 1). Twenty-one SNPs were transitions (mostly C/T), five were transversions, and one a combination. Ten of these SNPs (37%) were significantly associated with mortality rates (after Bonferroni correction), with three SNPs associated with a decreased mortality and six with an increased mortality. Four SNPs were located in non-structural proteins (nsps) from the Orf1ab gene, one in the S gene, one in the Orf8, three in the N gene, and one in the 3’Untranslated Region (UTR) (Table 1). The three SNPs from the N gene covered two consecutive codons with a change from AGGGG to AAACGA. All SNPs significantly associated with mortality rate caused changes in the amino acid sequence of the respective proteins, except C/T 2983 and C/T 8728 in Orf1ab, which were silent (Table 1).

We then elaborated new GLMs to test the association of SNPs combinations with patient mortality, again adjusting for patient age, sex, geographic region and month of the year, and models were compared based on Akaike information criteria. The best model included three SNPs that were significantly associated with patient mortality: C/T 2983 and T/C 14,353 in Orf1ab, and the 28,827–28,829 codons of the N gene (Table 2). These results indicate that Orf1ab and N viral proteins are key proteins which variants are associated with patient mortality, and to a lesser extent, Orf8 and the S glycoprotein.

To validate these results, we examined SARS-CoV-2 genomes from a second independent cohort of 992 patients (Appendix A). We tested nine SNPs identified in Table 1 and seven of these were found significantly associated with patient mortality (Table 3), mostly validating our initial results. The best GLM testing SNPs combinations included SNPs A/G 23349, C/T 14353 and CA/GG 28827–28828, as well as geographic region, month of the year and patient age, but not sex (Appendix A), providing further support that SNPs in Orf1ab, S and N genes are associated with patient mortality.

## 4. Discussion

We identified multiple SARS-CoV-2 genomic signatures in several viral genes that were associated with patient mortality in two independent cohorts. The functional significance of the variants identified here remains to be further investigated. Orf1ab encodes for several nsps, including the RNA-dependent RNA polymerase (RdRp), and it was previously identified as a mutation hot spot, suggestive of potential selection pressure associated with adaptation to human hosts [9]. The frequency of the T/C 14,353 variant (P4714L) has been found correlated with country mortality rates [15], but we found here that it was associated with a lower mortality. This variant falls within the RdRp and may affect viral replication. However, Orf1ab was also implicated in the pathogenesis of SARS-CoV-1 infections through processes distinct from viral replication that included cell signaling and the modulation of the immune response [18]. The proteins nsp2 and nsp3 have been proposed to play a role in COVID-19 pathogenesis [19], although SNP C/T 2983 associated with mortality and located in the nsp3 sequence did not cause a change in amino acid. Thus, this SNP may have an unknown function in addition to coding for nsp3. Orf8 from SARS-CoV-2 can interfere with type I interferon response in vitro [20], which has been found to be critical for mitigating disease severity [7]. The consequences of the R203K G204R substitutions in the N protein also warrant functional studies to assess its role in pathogenesis, as these are the most frequent variants in this protein [10]. Finally, it is interesting to note that the G614D substitution in the S protein was associated with an increased mortality rate in the bivariate analysis in both cohorts, and in the multivariate model of the validation cohort. Previous works showed that this substitution causes a greater infectivity and higher virus loads, but its effect on disease severity and mortality in patients has been debated [11,12,15]. Our data provide evidence that this substitution can lead to increased mortality.

The changes in the proportion of the different Clades we identified over time indicate that further monitoring is necessary. While changes in the proportion of variants over time can be expected due to founder effect of a virus rapidly spreading into naïve host populations, the associations of several of these variants with patient mortality may help better anticipate the risk for severe disease.

A limitation of our study is that the viral genomes which are sequenced may not be a random sample of the global virus population. Thus, these cohorts could be biased as sequencing effort may vary among health institutions, countries, and over time. Sequencing may also be biased according to patient status, and contact tracing may result in samples being epidemiologically linked. These potential biases may affect the proportion of genome variants and SNPs. The lack of standardized reporting of patient clinical status may also be a limitation and some patients may have died at a later time after sequences were reported. Finally, some comorbidities are known to increase the risk of severe disease, but could not be taken into account as these are not reported in these datasets. Nonetheless, variations in co-morbidities were in part taken into account in the analysis of mortality rates as we adjusted for geographic, age and temporal variations.

In conclusion, we identified here several previously undetected possible determinants of mortality in the SARS-CoV-2 genome. The identified SNPs are potential critical targets to assess the mechanisms underlying COVID-19 disease severity and warrant straightforward experimental validation in functional studies, and further confirmation in additional cohorts.

## Figures and Tables

**Figure 1 viruses-13-00227-f001:**
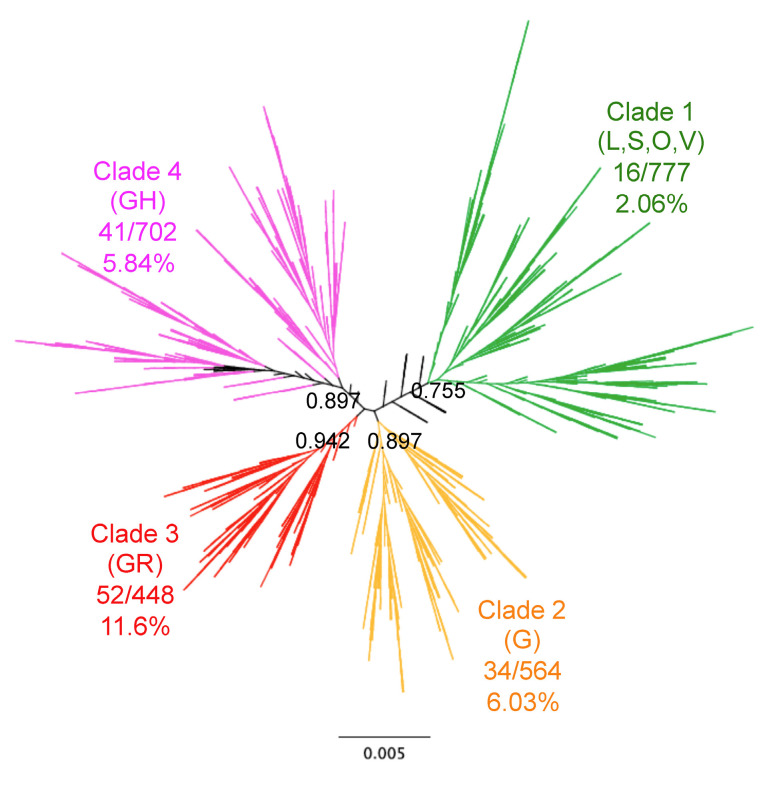
Genomic diversity of SARS-CoV-2 and patient mortality. Phylogenetic analysis of SARS-CoV-2 genomes from 2508 patients. The unrooted tree showed four main Clades (Clades 1–4) with a strong phylogenetic support as specified for each major branch, and mortality rates varied significantly among Clades as indicated (X^2^ = 47.93, d.f. =3, *p* < 0.0001). GISAID clade names are indicated in parenthesis for comparison.

**Figure 2 viruses-13-00227-f002:**
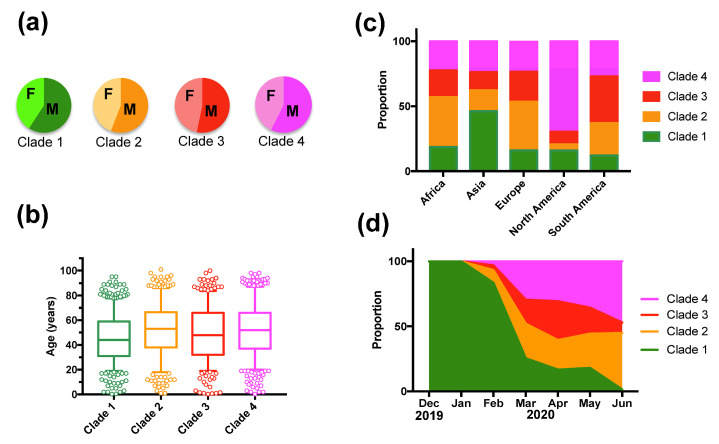
Patient sex-ratio, geographic and temporal distribution of SARS-CoV-2 clades. (**a**) There were no significant variations in patient sex-ratio among clades (X^2^ = 4.02, d.f. = 3, *p* = 0.26). (**b**) Patient age distribution among Clades was similar, although patients from Clade 1 were significantly younger (Tukey post hoc test, *p* < 0.01). (**c**) Geographic regions presented significant differences in Clade proportion (X^2^ = 522.65, d.f. = 12, *p* < 0.0001). (**d**) The proportion of Clades varied significantly over time (X^2^ = 821.54, d.f. = 18, *p* < 0.0001).

**Figure 3 viruses-13-00227-f003:**
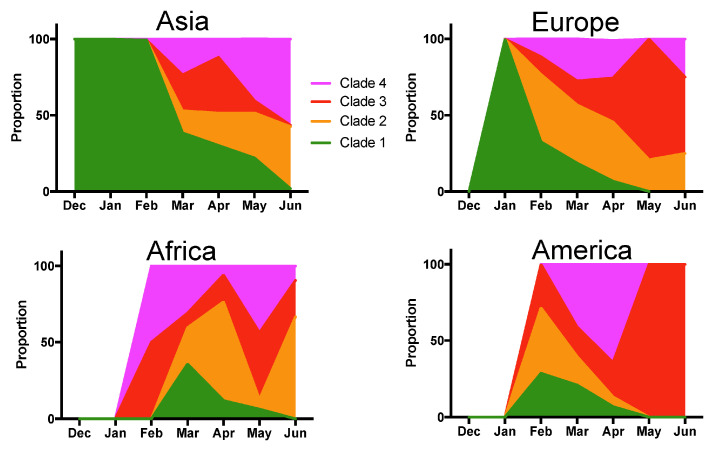
Temporal variations in Clade proportion per geographic region. Changes in the proportion of the indicated clades over time are shown for Asia, Europe, Africa and America.

**Table 1 viruses-13-00227-t001:** SNPs of SARS-CoV-2 and their association with patient mortality.

SNP	Position	Reference	Variant	X^2^	*p* Value *	OR	OR 95%CI	Protein	AA Change ^$^
C/T	1005	88.10%	11.90%	0.257	0.6125	1.15	0.66–1.99	nsp2	T265I
C/T	2362	97.60%	2.40%	2.602	0.1068	3.69	0.51–26.81	nsp2	No change
C/T	2782	96.10%	3.90%	1.852	0.173	0.59	0.29–1.2	nsp3	No change
C/T	2983	67.70%	32.30%	20.454	<0.0001 *	0.38	0.24–0.61	nsp3	No change
C/A	6258	95.80%	4.20%	0.177	0.674	0.82	0.33–2.06	nsp3	T2016K
C/T	8728	90.90%	9.10%	9.768	0.0018 *	3.67	1.35–10.1	nsp4	No change
G/T	11,029	87.60%	12.40%	11.947	0.0005 *	3.36	1.47–7.69	nsp6	L3606F
C/T	13,676	95.90%	4.10%	0.001	0.981	0.99	0.42–2.3	nsp12 (RdRp)	A4489V
T/C	14,353	67.40%	32.60%	40.812	<0.0001 *	0.22	0.13–0.39	nsp12 (RdRp)	P4714L
C/T	14,751	96.80%	3.20%	2.075	0.149	2.45	0.6–10.1	nsp12 (RdRp)	No change
C/T	15,270	94.10%	5.90%	3.381	0.066	0.56	0.310.99	nsp12 (RdRp)	No change
C/T	18,823	87.30%	12.70%	1.377	0.241	0.75	0.47–1.19	nsp14	No change
A/G	20,214	93.10%	6.90%	0.112	0.738	1.12	0.56–2.24	nsp15	No change
C/T	22,390	94.80%	5.20%	4.201	0.04	0.51	0.28–0.93	S	No change
A/G	23,349	31.90%	68.10%	38.338	<0.0001 *	4.23	2.46–7.27	S	G614D
C/T	23,875	96.00%	4.00%	0.109	0.741	1.16	0.47–2.91	S	No change
G/T	25,508	72.30%	27.70%	0.156	0.693	0.93	0.64–1.35	Orf3a	Q57H
G/T	26,090	95.90%	4.10%	6.572	0.0104	6.37	0.88–45.96	Orf3a	G251V
C/T	26,681	91.00%	9.00%	1.487	0.222	0.71	0.42–1.2	M	No change
T/C	28,090	90.20%	9.80%	11.194	0.0008 *	0.25	0.09–0.69	Orf8	L84S
C/T	28,257	95.70%	4.30%	0.199	0.656	1.22	0.49–3.05	N	P13L
C/T	28,800	94.70%	5.30%	2.587	0.108	0.58	0.31–1.08	N	S194L
G/A	28,823	97.70%	2.30%	6.812	0.0091	0	-	N	S202N
G/A ^#^	28,827	82.00%	18.00%	31.579	<0.0001 *	2.93	2.05–4.19	N	R203K G204R
G/A ^#^	28,828	82.00%	18.00%	32.172	<0.0001 *	2.96	2.08–4.24	N	R203K G204R
G/C ^#^	28,829	82.00%	18.00%	31.444	<0.0001 *	2.92	2.05–4.18	N	R203K G204R
G/A/T	29,688	95.60%	2.50%	12.896	0.0016 *	-	-	3’UTR	-

* indicate statistically significant association of Single-Nucleotide Polymorphisms (SNP) with mortality after Bonferroni correction (The adjusted threshold for significance was 0.00185). ^#^ These SNPs occur in the same sequences and affect two consecutive codons resulting in two amino acid changes. ^$^ Amino acid (AA) position within the respective proteins is indicated.

**Table 2 viruses-13-00227-t002:** Parameter estimates of the Generalized Linear Model (GLM) for patient mortality.

Term	Estimate	Std Error	X^2^	*p* Value	Lower CL	Upper CL
Intercept	−5.103	1.143	37.51	<0.0001 *	−7.533	−3.310
Africa	−1.069	0.675	5.37	0.021 *	−2.836	0.030
Asia	0.198	0.229	3.90	0.048 *	−0.227	0.717
Europe	−0.730	0.273	6.87	0.009 *	−1.264	−0.148
North America	0.239	0.282	3.31	0.069	−0.305	0.834
Jan	−1.929	1.265	2.08	0.149	−6.105	−0.096
Feb	−1.360	0.773	2.08	0.149	−3.292	0.010
Mar	−0.100	0.385	1.80	0.18	−0.766	0.818
Apr	0.810	0.379	8.76	0.003 *	0.161	1.721
May	0.317	0.454	2.16	0.142	−0.541	1.320
Jun	1.161	0.439	10.10	0.002 *	0.362	2.147
Female sex	−0.216	0.100	9.26	0.002 *	−0.419	−0.019
Age	0.052	0.006	108.63	<0.0001 *	0.041	0.064
C2983	1.833	0.753	11.03	0.001 *	0.690	3.585
T2983	−1.160	0.695	0.15	0.697	−2.147	0.530
C14353	−2.349	0.690	0.36	0.549	−3.514	−0.583
T14353	0.767	0.587	4.77	0.029 *	−0.107	2.434
AAC28827	0.001	0.578	1.07	0.301	−0.820	1.648
GGG28827	−0.824	0.573	0.41	0.523	−1.625	0.819

* Statistically significant *p* values. Std Error: standard error. CL: 95% confidence interval of estimates. The overall model had a X^2^ = 201.02, *p* < 0.0001. AICc = 779.7, with significant effect tests for geographic region (X^2^ = 54.27, d.f. = 4, *p* < 0.0001), time of year (X^2^ = 41.48, d.f. = 6, *p* < 0.0001, sex (X^2^ = 9.26, d.f. = 1, *p* = 0.002), age (X^2^ = 108.63, d.f. = 1, *p* < 0.0001), C/T2983 (X^2^ = 30.04, d.f. = 2, *p* < 0001), C/T14353 (X^2^ = 29.30, d.f. = 2, *p* < 0.0001, and CCA/GGG28827 (X^2^ = 14.96, d.f. = 2, *p* = 0.0006).

**Table 3 viruses-13-00227-t003:** SNPs of SARS-CoV-2 and their association with patient mortality in the validation cohort.

SNP	Position	Reference	Variant	X^2^	*p* Value *	Protein
C/T	2983	75.73%	24.27%	20.318	<0.0001 *	nsp3
C/T	8728	89.59%	10.41%	8.250	0.0041 *	nsp4
G/T/A	11,029	90.07%	9.93%	7.633	0.022	nsp6
T/C	14,353	73.71%	26.29%	10.845	0.0010 *	nsp12 (RdRp)
A/G	23,349	23.91%	76.09%	20.529	<0.0001 *	S
T/C	28,090	89.53%	10.47%	8.303	0.0040 *	Orf8
G/A ^#^	28,827	60.96%	39.04%	9.395	0.0022 *	N
G/A ^#^	28,828	60.99%	39.01%	9.374	0.0022 *	N
G/C/A ^#^	28,829	60.44%	38.94%	31.444	0.0094	N

* indicate statistically significant association of SNP with mortality after Bonferroni correction (The adjusted threshold for significance was 0.0055). ^#^ These SNPs mostly occur in the same sequences but not exclusively as in the first cohort.

## Data Availability

The data presented in this study are available in Appendix A.

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
