# Peer review of "Genomic Signatures of SARS-CoV-2 Associated with Patient Mortality"

_viruses, 2021, doi:10.3390/v13020227_

Round 1

Reviewer 1 Report

viruses-1041984

The authors tested for associations of SARS-CoV-2 viral genomic variants with fatal infection outcome from a small of 2,508 genomes and metadata taken from the GISAID data base.  Unfortunately the authors used a convenience set of 2508 samples for which the sequence and fatality outcome data were publicly available in the GISAID database.  The sample set of 2508 is tiny compared to the currently available genome number (>290,000 as of 25 Dec 2020). Conclusions based on such a selected and possibly biased sample set should not be reported as valid in the scientific literature and can lead to misinformation and false conclusions about SARS-CoV-2 and the possible outcome of SARS-CoV-2  infections.

The following points should be considered by the author.

  1. Documentation of the genomes with available outcome compared to all genomes should be provided to exclude for bias at this level. Documentation should be provided that the sampling and reporting practices for the 2508 included genomes are sufficiently uniform to support such an analysis should be provided.
  1. I don't find anywhere a listing of the mortality data and how these were reported the different groups. One confounding element is that frequently mortality is not easy to link to diagnostic samples. These mortality data should be listed in a table including the original reporting from GISAID and how the authors converted these to a common format.
  1. The metrics of how many different studies have contributed to the data set should be provided. How many total studies are presented in GISAID and how many studies report infection outcome (mortality vs survival). The authors should provide assurance that reporting practices between the different groups reporting data are consistent enough to allow using only 2508 genomes for an analysis. For example in our own SARS-CoV-2 sequencing work it is difficult to obtain follow-up and infection outcome data for every sample. Unless the sample was obtained from a deceased individual, obtaining a later outcome is frequently not possible. Such an analysis based on such a selective data set is unlikely to produce biologically relevant conclusions.
  1. Line 75: "Because many of the clinical terms used to describe disease status were ambiguous/vague and to avoid any bias in categorizing disease severity, we only focused on mortality v.s. survival (case fatality rate)." The authors should include in their complete data table all mortality/survival listings and the final mortality/survival conclusion used for the study.
  1. Line 101. "Sequences from Oceania were excluded from this analysis due to a small sample size of sequences from this region (15 sequences), which caused model instability. " Define "modal instability" The authors should provide some justification for this exclusion due to "small sample size" yet inclusion of the remaining 2508 genomes (<1% of total genomes available) as valid.
  1. Line 103 "To identify genomic variants, Single Nucleotide Polymorphisms (SNPs) were called from SARS-CoV-2 genome alignment through Geneious 11 SNP/variant tool, and tested individually for association with mortality, using X2 tests and odd ratios (OR). Statistical significance was adjusted using Bonferroni correction to account for multiple testing. SNPs positions in the genome were determined based on a sequence from Wuhan, China from December 30, 2019 (Genbank #MT291827) as reference. We also assessed the phenotypic effect of each SNP in the corresponding viral protein."
  1. Throughout text one or both of the established lineage systems for SARS-CoV-2 (Nextclade: https://clades.nextstrain.org/ Pangolin https://cov-lineages.org/pangolin.html) should be used. It appears that the authors have created their own nomenclature and this makes it difficult/impossible to link to the established lineages for SARS-CoV-2 reported in the literature.

Examples of the use of a non-conventional nomenclature:

Line 125 "…However, patient mortality varied significantly among the identified clades (X2=47.93, d.f.=3, P<0.0001), ranging from 2.06% [95%CI 1.28-3.32] for Clade 1 up to 11.61% [95%CI 8.97-14.91] for Clade 3, with Clades 2 and 4 having intermediate mortality rates (6.03% [95%CI 4.35-8.31] and 5.84% [95%CI 4.34-7.83], respectively). These results suggested an association of viral clades with mortality. "

Line 143 : "Clade 1, associated with a lower mortality rate, was predominant in Asia, and Clade 4 was predominant in North America (Fig. 1D)."  

and Figure 1 , throughout.

  1. An additional column in Table 1 should list frequency of each variant in the entire available genomes (290,000 genomes as of 25 Dec 2020)
  1. Line 174 Table 1 "Phenotype" this is not correct, the authors I believe mean amino acid change. Phenotype cannot be ascribed without functional data.
  1. Line 175: "*indicate statistically significant association of SNP with mortality after Bonferroni correction (The adjusted threshold for significance was 0.00185)."

More details on the Bonferroni correction should be provided, I would assume that each nt of the ca. 30,000 nt genome represents an independent test so the final p value should survive a 30,000 nt x 2508 sample Bonferroni correction. The stated threshold of  0.00185 seems too high to survive multiplication by (30,000 x 2508). A complete description of the  Bonferroni correction applied and an explanation and justification of the correction values should be provided.

  1. Line 229: " A limitation of our study is that the viral genomes which are sequenced may not be a random sample of the global virus population. Thus, these cohorts could be biased as sequencing effort may vary among health institutions, countries, and over time. Sequencing may also be biased according  to patient status, and contact tracing may result in samples being epidemiologically linked. These potential biases may affect the proportion of genome variants and SNPs. Finally, some comorbidities are know to increase the risk of severe disease, but could not be taken into account as these are not reported in these datasets. Nonetheless, variations in co-morbidities were in part taken into account in the analysis of mortality rates as we adjusted for geographic, age and temporal variations. "

Additional biases not listed include: sequencing platform, non-uniform reporting of clinical features. These should also be listed.

  1. Line 237: " In conclusion, we identified here several previously undetected likely determinants of mortality in the SARS-CoV-2 genome. "

Not " likely determinants" at best, "speculative associations".  Provide some mechanism for how such changes could alter infection outcome.

Author Response

  1. Documentation of the genomes with available outcome compared to all genomes should be provided to exclude for bias at this level. Documentation should be provided that the sampling and reporting practices for the 2508 included genomes are sufficiently uniform to support such an analysis should be provided.

ANSWER: Our goal was not to evaluate the frequencies of viral clades/variants among all genome sequences available, nor did we aim to explore all viral diversity as this goes beyond the scope of this study. Rather, our aim was to detect potential associations between viral genetic diversity and patient mortality in a specific cohort. Nonetheless, the cohort of patients/sequences used is large enough to cover most of the virus diversity and geographic origin, in spite of the potential limitations that are clearly acknowledged in the discussion (Page 9, lines 273-282).

2. I don't find anywhere a listing of the mortality data and how these were reported the different groups. One confounding element is that frequently mortality is not easy to link to diagnostic samples. These mortality data should be listed in a table including the original reporting from GISAID and how the authors converted these to a common format.

ANSWER: We have added details in the methods to better explain how clinical status is reported in the GISAID database and was interpreted in our study: We considered patients described as Deceased as dead, while those described as “Alive, Asymptomatic, Cured, Discharged, Discharged after recovery, Facility quarantine, Fever, Home, Hospitalized, ICU, Severe, In Hospital, Inpatient, Isolation, Live, Mild, Moderate, Outpatient, Not Hospitalized, Quarantine(d), Recovered, Released, Symptomatic” as survivors (Page 3, lines 81-84).

3. The metrics of how many different studies have contributed to the data set should be provided. How many total studies are presented in GISAID and how many studies report infection outcome (mortality vs survival). The authors should provide assurance that reporting practices between the different groups reporting data are consistent enough to allow using only 2508 genomes for an analysis. For example in our own SARS-CoV-2 sequencing work it is difficult to obtain follow-up and infection outcome data for every sample. Unless the sample was obtained from a deceased individual, obtaining a later outcome is frequently not possible. Such an analysis based on such a selective data set is unlikely to produce biologically relevant conclusions.

ANSWER: It is difficult to determine the number of “studies” that contributed the sequences used here, but the details of the source of each sequence are provided in the Supplementary Table 1 and 4, which indicate that at least 402 laboratories from around the world contributed sequences. As detailed above, due to difference in reporting practices regarding clinical outcome, we only considered mortality to be reliable and consistent enough for our analysis.

4. Line 75: "Because many of the clinical terms used to describe disease status were ambiguous/vague and to avoid any bias in categorizing disease severity, we only focused on mortality v.s. survival (case fatality rate)." The authors should include in their complete data table all mortality/survival listings and the final mortality/survival conclusion used for the study.

ANSWER: We have added some details in the methods to better explain how mortality is reported in the GISAID database and was interpreted in our study:

We considered patients described as Deceased as dead, while those described as “Alive, Asymptomatic, Cured, Discharged, Discharged after recovery, Facility quarantine, Fever, Home, Hospitalized, ICU, Severe, In Hospital, Inpatient, Isolation, Live, Mild, Moderate, Outpatient, Not Hospitalized, Quarantine(d), Recovered, Released, Symptomatic” as survivors (Page 3, lines 81-84).

5. Line 101. "Sequences from Oceania were excluded from this analysis due to a small sample size of sequences from this region (15 sequences), which caused model instability. " Define "modal instability" The authors should provide some justification for this exclusion due to "small sample size" yet inclusion of the remaining 2508 genomes (<1% of total genomes available) as valid.

ANSWER: The number of sequences for the different regions was large enough (several hundreds of sequences) to provide reliable estimates of mortality (as well as other parameters) for each region, while for Oceania and only 15 sequences, the estimates were very poor and not reliable due to such a small number of sequences. Including data from Oceania resulted in a meaningless model with extremely large confidence intervals, which is commonly referred to as an unstable model.

6. Line 103 "To identify genomic variants, Single Nucleotide Polymorphisms (SNPs) were called from SARS-CoV-2 genome alignment through Geneious 11 SNP/variant tool, and tested individually for association with mortality, using X2 tests and odd ratios (OR). Statistical significance was adjusted using Bonferroni correction to account for multiple testing. SNPs positions in the genome were determined based on a sequence from Wuhan, China from December 30, 2019 (Genbank #MT291827) as reference. We also assessed the phenotypic effect of each SNP in the corresponding viral protein." 

Throughout text one or both of the established lineage systems for SARS-CoV-2 (Nextclade: https://clades.nextstrain.org/ Pangolin https://cov-lineages.org/pangolin.html) should be used. It appears that the authors have created their own nomenclature and this makes it difficult/impossible to link to the established lineages for SARS-CoV-2 reported in the literature.

Examples of the use of a non-conventional nomenclature:

Line 125 "…However, patient mortality varied significantly among the identified clades (X2=47.93, d.f.=3, P<0.0001), ranging from 2.06% [95%CI 1.28-3.32] for Clade 1 up to 11.61% [95%CI 8.97-14.91] for Clade 3, with Clades 2 and 4 having intermediate mortality rates (6.03% [95%CI 4.35-8.31] and 5.84% [95%CI 4.34-7.83], respectively). These results suggested an association of viral clades with mortality. "

Line 143 : "Clade 1, associated with a lower mortality rate, was predominant in Asia, and Clade 4 was predominant in North America (Fig. 1D)."  

and Figure 1 , throughout.

ANSWER: We are aware of the different nomenclatures that have been proposed to identify SARS-CoV2 clades and lineages. As suggested, we have added the correspondence between our clades and those used in the GISAID database for easier comparison and both are now indicated in Figure 1.

7. An additional column in Table 1 should list frequency of each variant in the entire available genomes (290,000 genomes as of 25 Dec 2020)

ANSWER: As mentioned above, we believe that it is beyond the scope of the present study to evaluate the frequency of the different variants among the entire database of SARS-CoV2 sequences.

8. Line 174 Table 1 "Phenotype" this is not correct, the authors I believe mean amino acid change. Phenotype cannot be ascribed without functional data.

ANSWER: We agree with the reviewer and “Amino Acid Change” is now indicated.

9. Line 175: "*indicate statistically significant association of SNP with mortality after Bonferroni correction (The adjusted threshold for significance was 0.00185)."

More details on the Bonferroni correction should be provided, I would assume that each nt of the ca. 30,000 nt genome represents an independent test so the final p value should survive a 30,000 nt x 2508 sample Bonferroni correction. The stated threshold of  0.00185 seems too high to survive multiplication by (30,000 x 2508). A complete description of the  Bonferroni correction applied and an explanation and justification of the correction values should be provided.

ANSWER: Bonferroni correction uses a corrected significance level of a/nα / m {\displaystyle \alpha /m} , where α {\displaystyle \alpha } a is the desired significance level and nm {\displaystyle m} is the number of hypotheses tested, which in this case is the number of SNPs tested.

10. Line 229: " A limitation of our study is that the viral genomes which are sequenced may not be a random sample of the global virus population. Thus, these cohorts could be biased as sequencing effort may vary among health institutions, countries, and over time. Sequencing may also be biased according  to patient status, and contact tracing may result in samples being epidemiologically linked. These potential biases may affect the proportion of genome variants and SNPs. Finally, some comorbidities are know to increase the risk of severe disease, but could not be taken into account as these are not reported in these datasets. Nonetheless, variations in co-morbidities were in part taken into account in the analysis of mortality rates as we adjusted for geographic, age and temporal variations. "

Additional biases not listed include: sequencing platform, non-uniform reporting of clinical features. These should also be listed.

ANSWER: We disagree with the reviewer that sequencing platforms generate bias in the sequences obtained, as this would make all the sequence data of very limited usefulness. Although mortality is a simple and uniform criteria, some patients may indeed have died at a later time after sequences were reported, and we have thus added the non-uniform reporting of clinical status as an additional limitation as suggested (Page 9, lines 277-278). 

11. Line 237: " In conclusion, we identified here several previously undetected likely determinants of mortality in the SARS-CoV-2 genome. "

Not " likely determinants" at best, "speculative associations".  Provide some mechanism for how such changes could alter infection outcome.

ANSWER: We believe that our conclusion is well supported by our data, but have nonetheless reworded our conclusion to be more cautious, as suggested. It now reads: “we identified here several previously undetected possible determinants of mortality in the SARS-CoV-2 genome” (page 9, line 283). We do discuss some of the potential mechanisms underlying the association of the identified variants with mortality.

Reviewer 2 Report

Comments to manuscript by Dumonteil et al.  (041984) submitted to viruses 

First of all, I appreciate efforts of authors to collect many SARS-CoV-2 data and to conduct a series of statistical analyses. Generally speaking, this manuscript, after revision, should be published considering its importance to the global human health.

I have one major comment on their phylogenetic analysis and another major comment on significance of synonymous (no amino acid change) SNP at position 2983.

  1. In molecular phylogenetics, “strong phylogenetic support” usually means more than 95%, however, all phylogenetic supports of four clades are lower than 0.95. Therefore, if authors would like to stick to these four clades, they can eliminate some sequences near the central position, which is presumably close to the origin of pandemic or the common ancestor of these four clades. I think definition of these four clades is quite important for this paper, so the phylogenetic tree shown in Figure 1A may be shown as independent figure, not one panel of Figure 1.
  2. It is interesting that synonymous  (no amino acid change) SNP at position 2983 showed a very low P values in Tables 1, 2 and 3. Authors admitted at lines 213-214 of page 8: although SNP C/T 2983 associated with mortality and located in the nsp3 sequence did not cause a change in amino acid. One possibility is the statistical significance is false positive. This possibility is related to limitation of this study written in one paragraph in lines 229-236 of page 8. Another interesting possibility is the short nucleotide sequence including position 2983 may have some unknown function addition to coding protein nsp3. Biologically this possibility is quite inreresting, and I hope authors may add some more discussion on this point.

I have following minor comments.

  1. In Abstract, authors described cohort of 2,508 patients as “large”. However, in public health study in general, often more than 10,000 individuals are examined in cohort studies. Therefore, this word may be modified to some other word.
  2. In Figure 1A, authors showed an unrooted tree of SARS-CoV-2 sequences. It is better to add some appropriate outgrip sequence(s), and made this tree to be “rooted”. I suspect that black colored sequences may be outgroup sequences.
  3. There are two panel (b)’s in Figure 1. The later one should be labeled as (c).
  4. At line 220 of page 8: “Previous work” may be written in plural form.
  5. At line 222 of page 8: “can led to” may be written as “can lead to”.
  6. Supplementary Figure 1 is interesting, and this may be moved to main figure.

Author Response

First of all, I appreciate efforts of authors to collect many SARS-CoV-2 data and to conduct a series of statistical analyses. Generally speaking, this manuscript, after revision, should be published considering its importance to the global human health.

ANSWER: We thank the reviewer for his/her appreciation of our study.

I have one major comment on their phylogenetic analysis and another major comment on significance of synonymous (no amino acid change) SNP at position 2983.

  1. In molecular phylogenetics, “strong phylogenetic support” usually means more than 95%, however, all phylogenetic supports of four clades are lower than 0.95. Therefore, if authors would like to stick to these four clades, they can eliminate some sequences near the central position, which is presumably close to the origin of pandemic or the common ancestor of these four clades. I think definition of these four clades is quite important for this paper, so the phylogenetic tree shown in Figure 1A may be shown as independent figure, not one panel of Figure 1.

ANSWER: We agree with the reviewer and have reworded our statement to refer to “good phylogenetic support” as it is indeed in the range of 75-94%. We prefer not to eliminate any sequences from our analysis as this would seem arbitrary and may bias the analysis. Also, we now present the phylogenetic tree as an independent figure as suggested (Figure 1).

2. It is interesting that synonymous  (no amino acid change) SNP at position 2983 showed a very low P values in Tables 1, 2 and 3. Authors admitted at lines 213-214 of page 8: although SNP C/T 2983 associated with mortality and located in the nsp3 sequence did not cause a change in amino acid. One possibility is the statistical significance is false positive. This possibility is related to limitation of this study written in one paragraph in lines 229-236 of page 8. Another interesting possibility is the short nucleotide sequence including position 2983 may have some unknown function addition to coding protein nsp3. Biologically this possibility is quite inreresting, and I hope authors may add some more discussion on this point.

 ANSWER: We agree with the reviewer that this SNP may have an unknown function in addition coding for NSP3, and this has been added to the discussion as suggested (Page 11, lines 249-250).

I have following minor comments.

  1. In Abstract, authors described cohort of 2,508 patients as “large”. However, in public health study in general, often more than 10,000 individuals are examined in cohort studies. Therefore, this word may be modified to some other word.

ANSWER: We have deleted “large” as suggested by the reviewer (Page 1, line 16).

2. In Figure 1A, authors showed an unrooted tree of SARS-CoV-2 sequences. It is better to add some appropriate outgrip sequence(s), and made this tree to be “rooted”. I suspect that black colored sequences may be outgroup sequences.

ANSWER: We do not focus on the evolutionary history of the virus and merely aimed to illustrate the sequence diversity from our cohort, which is why an unrooted tree is shown.

3. There are two panel (b)’s in Figure 1. The later one should be labeled as (c).

ANSWER: Figure 1 has now been split into two figures, which have been relabeled accordingly.

4. At line 220 of page 8: “Previous work” may be written in plural form.

ANSWER: Spelling has been corrected as suggested.

5. At line 222 of page 8: “can led to” may be written as “can lead to”.

ANSWER: Spelling has been corrected as suggested.

6. Supplementary Figure 1 is interesting, and this may be moved to main figure.

ANSWER: This figure has been moved as a main figure (Figure 3) as suggested.

Reviewer 3 Report

Dumonteil et al deal with a very important topic - can one relate disease outcome to the genomic sequence of the virus? The answer to this question has obvious medical benefits. The authors use a statistically sound cohort of data to analyze the information. In general their analysis is sound and the only shortcomings in it are inherent, but nonetheless important.

The issue of co-morbidity is critical. The authors specify that this is the case in their introduction, but due to lack of information in their data base it was neglected. Are other data sets available that include this information? The authors should at the very least model what is the potential influence of co-morbidity on their results. Would it render their results irrelevant?

An additional point is protein versus nucleotide sequence. It would be very helpful if table 1 would contain the amino acid change. It may also contain the conservation fo that amino acid as a further bit of data to understand the potential important of the SNP

Author Response

Dumonteil et al deal with a very important topic - can one relate disease outcome to the genomic sequence of the virus? The answer to this question has obvious medical benefits. The authors use a statistically sound cohort of data to analyze the information. In general their analysis is sound and the only shortcomings in it are inherent, but nonetheless important.

ANSWER: We thank the reviewer for his/her appreciation of our study.

The issue of co-morbidity is critical. The authors specify that this is the case in their introduction, but due to lack of information in their data base it was neglected. Are other data sets available that include this information? The authors should at the very least model what is the potential influence of co-morbidity on their results. Would it render their results irrelevant?

ANSWER: We agree with the reviewer that co-morbidities are important, but as we pointed out these could not be taken into account because they are absent from this database. We are unaware of other public databases that include this information, but greatly appreciate the reviewer's comment and are working to build such a database moving forward based on this helpful suggestion. We also hope that our work would spur future studies that may address this point. 

An additional point is protein versus nucleotide sequence. It would be very helpful if table 1 would contain the amino acid change. It may also contain the conservation fo that amino acid as a further bit of data to understand the potential important of the SNP

ANSWER: Amino acid changes are indicated in the last column of Table 1.

Round 2

Reviewer 3 Report

I thank the authors for addressing my question/concern. I would recommend accepting the manuscript as is.